# OpenReview forum: "Bob’s Confetti: Phonetic Memorization Attacks in Music and Video Generation"
_ICLR.cc/2026/Conference — ICLR 2026 Conference Withdrawn Submission_

### Official Review · Reviewer_Ab5e · 2025-10-27

**Soundness:** 3
**Presentation:** 3
**Contribution:** 2
**Rating:** 4
**Confidence:** 4

**Summary:**

This paper introduces Adversarial PhoneTic Prompting (APT), a new method to bypass copyright filters in generative models. By using homophonic but semantically different lyrics (e.g., "mom's spaghetti" -> "Bob's confetti"), the authors show that L2S models (SUNO, YuE) regenerate copyrighted melodies. More surprisingly, this 'phonetic-to-visual leakage' is shown in a T2V model (Veo 3) reconstructing original video scenes from only the APT lyrics. The paper argues models memorize phonetic structure, not just text, exposing a key vulnerability.

**Strengths:**

* The APT concept is highly novel. It exposes a fundamental flaw in current text-based filters. The 'phonetic-to-visual leakage' finding is particularly surprising and significant.
* The paper is very clear, using good examples. The phonetic matching method (using CMUdict) is sound.

**Weaknesses:**

* The results are based on a small (N ~= 30), curated set of 'iconic' songs (e.g., "Lose Yourself"). It's unclear if this is a general vulnerability or just extreme overfitting on these specific tracks.
* The 'phonetic-to-visual' claim rests on a single N=1 case study ("Lose Yourself"). This is compelling but anecdotal, not generalizable proof.
* The paper only shows that the attack works, not why. It relies on output-level metrics (AudioJudge, MIRA) and lacks analysis of the model's internal workings (e.g., in an open-source model like YuE).

**Questions:**

1. How do you prove this is a general phonetic vulnerability, not just an artifact of overfitting on the few 'iconic' songs you tested?
2. Were you able to replicate the visual regurgitation on any other songs besides "Lose Yourself" on T2V?
3. Did you investigate the internal mechanism? For example, in YuE, does the APT prompt activate the same cross-attention patterns as the AVT (verbatim) prompt? Have you tried ablating the neurons responsible for AVT memorization to see if it also breaks the APT attack? This would provide very strong evidence.

---

### Official Review · Reviewer_zBm1 · 2025-10-31

**Soundness:** 2
**Presentation:** 3
**Contribution:** 2
**Rating:** 4
**Confidence:** 4

**Summary:**

This paper presents a novel class of attacks on generative music and video models, which exposes memorization vulnerabilities in text-conditioned lyrics to music systems such as SUNO, YuE. APT replaces original lyrics with homophonic but semantically unrelated phrases to bypass text-based copyright filters. The authors demonstrate that these models reproduce songs and even visual scenes that closely resemble the originals.

**Strengths:**

1. The concept of phonetic prompting as a trigger for memorization is novel and interesting.

2. The study evaluates multiple genres, languages, and models.

3. The paper is clearly written and easy to follow.

**Weaknesses:**

1. Since the structure and rhythm of the lyrics largely remain pretty similar after phonetic transformations, it seems natural and intuitive that the model would follow the original song, especially given that overfitting to training data is a well-known issue in generative models. Moreover, the songs selected for the experiments are all highly popular tracks. Therefore, it seems like the proposed attack is more like an attack for a text-based filter. This makes the contribution somewhat less surprising.

2. There is only one qualitative result for Veo 3 cross-modal evaluation, which makes the claims less convincing.

**Questions:**

1. Could the authors provide evidence that certain copyrighted tracks are indeed overrepresented in the training data, thereby amplifying this effect? Additionally, does the YuE model employ any form of text-based filtering similar to SUNO?

2. Have the authors explored simple countermeasures such as syllable- or rhythm-aware regularization to assess whether these techniques could mitigate phonetic-triggered leakage?

3. How were phonetic transformations generated for non-English languages such as Korean and Mandarin? (All with Claude 3.5-Haiku?). I'm asking this because after reviewing the project page, I noticed that for the Teresa Teng example, the so-called “transformed” lyrics appear identical to the original, which raises my main concerns about whether the APT process was correctly applied to non-English data. Can you provide more clarification on how multilingual transformations were constructed and validated?

4. How many human listeners are involved in the subjective study?

---

### Official Review · Reviewer_nErc · 2025-11-01

**Soundness:** 3
**Presentation:** 3
**Contribution:** 3
**Rating:** 6
**Confidence:** 4

**Summary:**

This paper presents Adversarial Phonetic Prompting (APT), demonstrating that phonetically similar but semantically distinct lyrics can bypass text-based filters in Lyrics-to-Song systems and trigger memorization of copyrighted content. The authors construct a composite phonetic similarity metric (Φ) using CMUdict features and evaluate APT's effectiveness across multiple L2S systems (SUNO, YuE) and languages, with a case study extending to Text-to-Video (Veo 3). Results show high melodic/rhythmic similarity to original songs (Tables 2, 3), measured via AudioJudge (LLM-based) and MiRA (CLAP/CoverID). The insight that sub-lexical phonetic patterns can serve as memorization keys is valuable and practically important, though the empirical validation has notable limitations.

**Strengths:**

The paper shifts attention from semantic/token-level memorization to phonetic cadence as a cross-modal retrieval cue. This is an underexplored angle with clear practical implications for content moderation. The Φ metric decomposes into interpretable CMUdict features; methodology is explicit and Figure 1 effectively communicates the concept. The paper demonstrates a real vulnerability in deployed commercial systems, motivating phonetic-aware defenses. The multilingual coverage (English/Chinese/Korean) strengthens generalizability claims. The Veo 3 case study (Fig. 4) is genuinely interesting and suggests memorization spans modalities, though this may needs more systematic validation.

**Weaknesses:**

(1) Tables 2-3 lack confidence intervals, variance estimates, or significance tests. How stable are results across random seeds? What's the variance in AudioJudge scores? These details are critical for a claim about "consistent" similarity.

(2) AudioJudge is LLM-based and designed by the authors. While human correlation exists (Fig. 7), the sample is small (~20 pairs). I'd like to see (a) larger human validation, (b) comparison with non-LLM metrics (pitch extraction + DTW, beat tracking), and (c) sensitivity analysis to judge prompts.

(3) The paper focus on iconic, culturally pervasive songs (Lose Yourself, Bohemian Rhapsody) where memorization is unsurprising. Would APT work on moderately popular or obscure copyrighted songs? This is crucial for understanding the threat model's scope.
Closed-system dependency: SUNO/Veo are proprietary; reproducibility requires open-source alternatives. The claim that SUNO doesn't filter Mandarin/Cantonese verbatim lyrics needs systematic documentation (sample size, false-negative rate).

(4) The Veo 3 evidence rests on qualitative frames from a handful of prompts. It'd be better if the authors can show quantitative visual similarity metrics and negative controls (songs where APT doesn't trigger visual regurgitation).

**Questions:**

(1) Can you provide statistical tests (paired t-tests, Wilcoxon) comparing APT vs. verbatim vs. random controls, with multiple generations per song?

(2) Please report AudioJudge inter-rater reliability and validate against a larger human panel (100+ song pairs) plus non-LLM baselines.

(3) What happens with less iconic songs? Please test on lesser-known songs with 10M-100M streams vs. >1B to understand memorization thresholds.

(4) For Veo 3: Please add quantitative metrics (such as pose/scene similarity, CLIP-based visual alignment) and report failure cases where APT doesn't trigger visual regurgitation.

(5) Will you release the Φ scorer and APT generation pipeline? Even without audio, this would enable testing on open models (e.g., MusicGen, Stable Audio). If so, under what license?

**Details Of Ethics Concerns:**

The authors provides a recipe for bypassing content filters. While the ethics statement is present, I recommend review focused on responsible disclosure protocols (e.g., vendor notification timeline) and mitigation artifacts (phonetic-aware filter prototypes), as APT could lower barriers to copyright circumvention.

---

### Official Review · Reviewer_QVie · 2025-11-03

**Soundness:** 2
**Presentation:** 2
**Contribution:** 1
**Rating:** 2
**Confidence:** 5

**Summary:**

This paper explores a phenomenon the authors call Adversarial Phonetic Prompting (APT), rewriting song lyrics into phonetic near-matches (like “Bob’s confetti” for “Mom’s spaghetti”) to see whether music generation models (Suno, YuE) still reproduce melodies or rhythms of specific copyrighted songs. The authors argue that such phonetic scaffolds could trigger memorized melodies even when direct lyrics are blocked. They also test “Adversarial Verbatim” (AVT) lyrics for baseline comparison and claim to find “cross-modal leakage,” where phonetic prompts also bias text-to-video models (Veo 3) toward similar visual scenes.

**Strengths:**

- The paper proposes a **fresh probing angle** for memorization: phonetic rather than textual.
- The idea is intuitively clever and potentially useful for understanding **cross-modal correlations** between language and sound.
- They report clear, reproducible prompting procedures (phoneme-based scoring, CMUdict alignment) and document case studies (e.g., “Lose Yourself”).
- As an exploratory piece, it raises valid questions about **model safety** and how phonetics might serve as a side channel for retrieval.

**Weaknesses:**

1. **Weak motivation.**
   The central claim—that platforms block verbatim lyrics and thus phonetic rewrites are a meaningful bypass—is *out of sync with real product behavior*. Suno does *not* block verbatim lyrics (I’ve generated the “Lose Yourself” song in Suno; it sounds similar but not identical), and the paper doesn’t evaluate Udio or other systems that actually enforce such filters. Also, SONICS (ICLR 2025) generates songs from Suno using real lyrics, which again sound similar but not identical. Without clear platform evidence, the motivation collapses.

2. **Misframed copyright risk.**
   The “cross-modal leakage” examples (e.g., hooded rapper, graffiti) are more like **stylistic similarities**, not plagiarism or literal reconstruction. If the resulting songs and videos are “similar but not the same,” this isn’t a meaningful legal or ethical breach—it’s just model style convergence. The paper overstates the risk without quantifying “substantial similarity” using musicological metrics.

3. **Problem not clearly important.**
   If platforms legally produce **AI covers** or user-generated remixes, identifying phonetic similarity provides no policy or practical benefit. The authors treat similarity itself as a threat but don’t demonstrate any real-world harm.

**Questions:**

Why this work is important, will benefit the society?

---

> ### Author Response · Authors · 2025-11-27
>
> We thank Reviewer QVie for the reviews and the questions. We believe that addressing to the reviewer's points provides the potential to strengthen our paper and below we respond to each concern in detail.
>
> **Response to Weakness 1 (*Weak Motivation*)**: Thank you very much for the review. We understand that the reviewer might have noticed that SUNO might not block by inputting verbatim lyrics from the original song. We also understand that previous work (SONICS from ICLR 2025) may have used verbatim lyrics to conduct experiments. However, we would first like to point out that SONICS was first released in February 2025 and during that time SUNO AI might not have had a strong copyright filter. During our experiment, we had encountered a very strong copyright filter where we were not able to generate any given song, which proves that our motivation still stands
>
> **Response to Weakness 2 (*Misframed copyright risk*)**: We appreciate the concern regarding legal framing. To clarify, our claim is not that APT outputs perfect replicas of the original content, but that the model exhibits substantial structural similarity driven by memorized phonetic alignment (not just style alone). Our experimental evaluation supports this difference where AudioJudge Similarity scores reach 0.90-0.98 in melody and rhythm for multiple APT prompts and MiRA CoverID values show overlap indicative of training data reuse rather than transformed creative generation. In addition, our human listening judgements reported the same effect. We will adjust the framing to emphasize that the core contribution is uncovering a sub-lexical memorization pathway, not claiming legal infringement.
>
> **Response to Weakness 3 (*Problem not clearly important*)**: We agree the framing should more clearly convey why this behavior matters beyond similarity for its own sake. The core problem statement of our paper is not that the mode produces perfect replicas, but that phonetic structure alone can trigger memorized musical and even visual content despite semantic distortion and active text-based copyright filters. This matters because existing safeguards assume that preventing exact lyrics or semantic matches is sufficient; however, our findings show that models like SUNO still generate outputs with high similarity scores (based on both quantitative results and qualitative demonstrations) and reproduce genre-specific visual features even when the input text is nonsensical. In other words, the practical risk is the current filtering and policy mechanisms are aligned to lexical similarity, while real leakage may occur through sub-lexical rhythmic and phonetic pathways that are currently unmonitored and unmitigated.
>
> **Response to Question 1 (*Importance*)**:
> - We understand that the reviewers are concerned about the significance of this paper especially due to the collapse of the motivation. However, we would like to emphasize that this work exposes a class of model behavior that current safety, copyright, and platform policies do not account for, even as generative music and multimodal systems rapidly transition into consumer facing products. Based on our observations, existing safeguards operate at the level of text matching or semantic similarity, yet our findings show that models can still reproduce copyrighted musical structure even when triggered by phonetic or rhythmic cues.
> - In addition, we believe that as generative media becomes integrated into creative tools, education platforms, streaming systems, and commercial licensing workflows, understanding these hidden leakage pathways is essential not only for preventing accidental copyright violations, but also for building transparent, accountable models that respect creators’ rights and avoid unintended replication of proprietary content. As we are the first to propose such a type of memorization could exist, identifying and measuring this behavior early allows our findings to serve as a foundation for more rigorous auditing methods, safer deployment policies, and future defenses that enable responsible and compliant use of generative media systems.

---

> > ### Comment · Reviewer_QVie · 2025-11-27
> >
> > > During our experiment, we had encountered a very strong copyright filter where we were not able to generate any given song, which proves that our motivation still stands
> >
> > I just generated "Lose Yourself" song **again** using Suno (Nov 27, 2025).  So I don't see the filter at all!
> >
> > For other sections I'm not convinced by the explanations. So, I'll keep my score.

---

### Note · Authors · 2026-01-07

I have read and agree with the venue's withdrawal policy on behalf of myself and my co-authors.